# Conditioning Regimens in Patients with β-Thalassemia Who Underwent Hematopoietic Stem Cell Transplantation: A Scoping Review

**DOI:** 10.3390/jcm11040907

**Published:** 2022-02-09

**Authors:** Olga Mulas, Brunella Mola, Giovanni Caocci, Giorgio La Nasa

**Affiliations:** Hematology Unit, Businco Hospital, Department of Medical Sciences and Public Health, University of Cagliari, 09124 Cagliari, Italy; mola.brunella@gmail.com (B.M.); giovanni.caocci@unica.it (G.C.); giorgio.lan@unica.it (G.L.N.)

**Keywords:** β-thalassemia major, allogenic hematopoietic stem cell transplantation, conditioning regimens

## Abstract

The success of transplant procedures in patients with beta-thalassemia major (β-thalassemia) goes hand-in-hand with improvements in disease knowledge, better supportive care, discoveries in immunogenetics, increase in stem cell sources, and enhancement of conditioning regimens. The aim of this scoping review was to report the evolution of conditioning regimes for β-thalassemia hematopoietic stem cell transplantation. We performed a systematic search for all relevant articles published before July 2021, using the following Medical Subject Headings: “bone marrow transplantation”, “stem cell transplantation”, “allogeneic”, “thalassemia”, “β-thalassemia”, and “thalassemia major”. The final analysis included 52 studies, published between 1988 and 2021, out of 3877 records. The most common conditioning regimen was a combination of busulfan and cyclophosphamide, with successive dose adjustments or remodulation based on patient characteristics. Pre-transplant treatments, reductions in cyclophosphamide dosage, or the adoption of novel agents such as treosulphan all improved overall survival and thalassemia-free survival in transplant-related mortality high-risk patients. Conditioning regimes were modulated for those without a suitable fully matched sibling or unrelated donor, with encouraging results. Hematopoietic stem cell transplantation with haploidentical donors is currently available to virtually all patients with β-thalassemia. However, disparities in outcome are still present around the world. In developing and limited-resource countries, where most diagnoses are focused, transplants are not always available. Therefore, more efforts are needed to close this treatment gap.

## 1. Introduction

Beta-thalassemia major (β-thalassemia) is a common monogenic disease characterized by abnormal hemoglobin structure [1]. Historically, β-thalassemia occurred predominantly in the region spanning sub-Saharan Africa, the Mediterranean, and the Middle East, stretching to south and southeast Asia [2]. However, global migration has spread β-thalassemia worldwide [3]. The most common mutations that cause β-thalassemia are single nucleotide substitutions, small deletions, or insertions within the β-globin gene [1]. These mutations reduce production of β-globin chains and HbA1. The degree of imbalance between α-globin and β-globin chains then causes accumulation of defective α-globin complexes that damages red blood cells [4]. This condition determines anemia severity, transfusion dependency, and overall clinical morbidity in β-thalassemia [5].

The first therapy developed for β-thalassemia was a combination of red blood cell transfusions and iron chelation that proved to be effective for efficient delivery. Insufficient iron chelation can cause tissue overload, leading to heart failure, liver fibrosis, and endocrine disorders [6]. However, adequate support substantially improves patient survival and quality of life [2]. New approaches have also been established to treat β-thalassemia, such as luspatercept, an inhibitor of ineffective erythropoiesis that acts as an activin IIB receptor-ligand trap [7]. Gene therapy is another proposed strategy, involving the infusion of autologous hematopoietic stem cells (HSCs) modified with a lentiviral vector that expresses β-globin in erythroid progenitors [8]. Despite the potential of this approach to completely cure β-thalassemia, cost management and long-term safety limit its clinical application [2,9].

Allogenic HSC transplantation (HSCT) has been proposed as a possible curative option since the 1980s [10]. Pediatric HSCT has better results than adult HSCT, and in 1990, the Pesaro risk score was proposed to assess transplant-related mortality in pediatric patients [11]. Patients were divided into three classes based on presence of hepatomegaly, portal fibrosis, and a history of inadequate iron chelation. Class 1 had no risk factors, class 2 had one or two risk factors, and class 3 had three risk factors. The 3-year overall survival (OS) for class 1 was 94% and dropped to 61% for class 3 [11]. Class 3 also contained a group of very high-risk (HR) patients, typically aged ≥ 7 years and with liver size ≥ 5 cm from the costal arch [12]. Unfortunately, a major complication of HSCT is acute and chronic graft-versus-host disease (GVHD), a potentially multi-systemic disorder caused by immunoeffector donor lymphocytes destroying host tissues; GVHD remains the main treatment-related cause of death in HSCT-treated β-thalassemia patients. However, constant advancements in our understanding and better management of transplant-related complications have improved outcomes over the years. Furthermore, clinicians now use different sources of stem cells and donors such as cord blood (CB) stem cells, peripheral blood stem cells (PBSC), matched unrelated donors (MUD), or haploidentical siblings even in non-oncological diseases such as β-thalassemia [2]. The evolution of treatment regimens and modulation of drug combinations or dosages have improved β-thalassemia HSCT outcomes.

This scoping review aimed to provide an overview of the conditioning regimens used in HSCT of β-thalassemia patients. We sought to describe the evolution of protocol regimes based on patient and donor characteristics, as well as on stem cell sources.

## 2. Methods

### 2.1. Search Strategy

The search identified journal articles published before July 2021 that describe conditioning regimens for HSCT in β-thalassemia patients. Articles were retrieved from PubMed, Scopus, and Web of Science. As keywords, we used Medical Subject Headings such as “bone marrow transplantation”, “stem cells transplantation”, “allogeneic”, “thalassemia”, “β-thalassemia”, “thalassemia major”, and variations thereof. A more extensive description of the search strategy is provided in Appendix A.

### 2.2. Inclusion and Exclusion Criteria

Studies were included if they were published in peer-reviewed journals, in English, examined more than four patients with β-thalassemia who underwent HSCT, described the protocols of conditioning regimens utilized in HSCT, and were not abstracts or reviews.

### 2.3. Article Selection

Two authors (O.M. and B.M.) selected articles for the review. Titles were first screened to eliminate irrelevant articles. Abstracts were then reviewed to confirm eligibility, and selected articles were processed for full-text analysis. The search also included studies cited in relevant articles.

PubMed, Scopus, and Web of Science yielded 3877 articles (Figure 1). After removing duplicates, 1842 remained. Of these, 1729 studies were excluded based on the title or abstract. After a full-text analysis of 113 potentially relevant articles, 61 were excluded: 11 for including fewer than four patients, 16 for being reviews or abstracts, 2 for not being written in English, 4 for not involving β-thalassemia patients, and 15 for the absence of a transplant procedure. In 13 studies, data could not be extracted from β-thalassemia patients alone. Overall, 52 patients were included in the final review.

### 2.4. Data Extraction

Two authors (O.M. and B.M.) tabulated data from each eligible study using a data charting form approved by all co-authors. When available, the following information was extracted: first author, year of publication, country of origin, number of patients, median age, sex, Pesaro risk score, conditioning regime protocol, GVHD prophylaxis, graft source, donor type, rate of acute and chronic GVHD, transplantation-related mortality (TMR), graft failure (GF), other complications, OS, thalassemia-free survival (TFS), and month follow-up.

## 3. Results

Table 1 summarizes the 52 studies included in this review. The selected studies ranged from the 1980s to 2021. Initial experiences with thalassemia patients undergoing allogeneic HSCT were limited to matched sibling donors (MSDs) who provided human leukocyte antigen (HLA)-compatible marrow stem cells. Busulfan (Bu) and cyclophosphamide (Cy) were clearly the backbone conditioning regimes in most studies. The first large study reported OS and TFS rates of 82% and 75%, respectively [11]. Over time, advances in pathology and developments in transplantation procedures improved outcomes, reducing TRM, graft failure, and GVHD incidence. Class 1 and class 2 patients had a probability of survival and TFS of 95–85% and 90–81%, respectively. When a classical dosage of Bu (3.5 mg/kg/day) and Cy (50 mg/kg/day) was administered, class 3 patients had worse OS, TFS, and TRM than class 1 patients [11]. A subsequent reduction in Cy to 30–40 mg/kg/day in class 3 patients improved their OS (78% vs. 61%) and reduced TRM (19% vs. 37.5%) [11,13]. Another study reported excellent OS and TMR (100% and 0%, respectively), acceptable rates of acute and chronic GVHD (11% and 8%), but high graft failure (20.6%), after treating 29 class 3 patients with a reduced intensity conditioning (RIC) regimen involving 4 mg/kg/day of Bu, 35 mg/m^2^/day of fludarabine (Flu), and one 500 cGy dose of total lymphoid irradiation (TLI) [14]. However, the use of TLI or more extensive total body irradiation (TBI) is not always associated with encouraging results in β-thalassemia patients, especially because of the high TRM rate [15,16,17,18].

A high graft failure rate requires a more active protocol for pretransplant immunosuppression (PTI) and myelosuppression. Class 3 patients experienced reduced TMR and graft failure, as well as increased OS and TFS, upon being treated with protocol 26 (P26), a pretransplant treatment involving 30 mg/kg/day of hydroxyurea (HU), 3 mg/kg/day of azathioprine (Azat), 20 mg/m^2^/day of Flu, and anti-thymocyte globulin (ATG) in addition to Bu–Cy [24,58]. The use of PTI was also extended in HR class 3 patients. When compared with class 3 patients who underwent Bu–Cy or analogous regimens, pretreatment with 40 mg/m^2^/day of Flu, 25 mg/m^2^/day of dexamethasone (DEX), plus 35 mg/m^2^/day of Flu and 130 mg/m^2^/day of Bu, resulted in similar rates of GVHD and TRM, but no graft failure [54]. Age at the time of transplantation is another factor that influences transplantation outcome, and changes made to the drug intake schedule have been insufficient to address the complicated effects [22,25,27,33,51,64]. A German study on β-thalassemia in children and adolescents receiving MSD-HSCT yielded encouraging results after completely replacing Cy with Flu; no GVHD cases or mortality were reported after 25 months of follow-up, and all patients were healthy [28].

The percentage of patients who can find a compatible donor within the family ranges from 25% to 30% [65]. For the remaining patients, HSCT from an MUD represents a viable option. A 2002 Italian study was the first to examine a large series of 32 β-thalassemia patients receiving MUD-HSCT, with a combination of Bu–Cy and other drugs (e.g., TTP) administered as conditioning regimens. Their OS and TFS were 79% and 66%, respectively; acute and chronic GVHD incidences was high, with a rejection rate of 12.5% and TRM rate of 19% [22]. The same research group later obtained better results in class 1 and 2 patients (OS, 96.7%; TFS, 80%) than in class 3 patients (OS 65.5%, TFS 54.5%) through increasing the accuracy of MUD detection using high-resolution HLA molecular typing and replacing Cy with Flu in selected patients [27]. Another study found that the Bu–Cy regimen in MUD-HSCT did not result in significantly worse OS and TFS than MFD-HSCT plus Bu–Cy combined with Flu and TLI [30]. Although the follow-up period was short, improved outcomes were also obtained in class 3 patients receiving MUD-HSCT along with a conditioning regimen of PTI using Flu (150 mg/m^2^/day), Cy (1 g/m^2^/day), and DEX (20 mg/m^2^/day) [63]. When Bu was replaced with treosulphan (Treo) in a Bu–Cy-based regimen, clinical outcomes in HR class 3 patients improved. Indeed, a Treo (14 g/m^2^/day), Flu (30 mg/m^2^/day), and TTP (8 mg/kg) protocol was associated with significantly lower TRM incidence (13% vs. 46%) and sinusoidal obstruction syndrome or veno-occlusive disease (VOD) (30% to 78%), when compared with Bu–Cy. In addition, OS (86.6% vs. 39.4%) and TFS (77.8% vs. 32.4%) were significantly better in the Treo group [51]. A study examining 20 patients found no significant differences in TFS between classes 1 and 2 vs. class 3 after a similar Treo-based regimen and BM-HSCT [35]. A larger subset of 60 patients treated with a Treo-based regimen resulted in a 5-year OS and TFS of 93% and 84%, respectively, with a 78% TFS probability for MUD recipients [48]. However, the same protocol in an Indian cohort was less encouraging, with a reduction in OS and TFS [50].

Despite a very high risk of GVHD and graft failure, non-BM stem cells (e.g., PBSC and CB stem cells) have been used in transplantation treatments for β-thalassemia. One of the earliest published studies on PBSC transplantation examined a small cohort treated with a Bu–Cy-based regimen and emphasized the importance of T cell depletion in this setting [29]. However, a comparison between PBSC and BM as an HSC source revealed that PBSC increased the risk of acute and chronic GVHD (72% and 48% for PBSC vs. 55% and 19% for BM), without any differences in 2-year OS or TFS [36]. Another study compared the outcomes of 52 children undergoing MUD-PBSC transplantation with those of 30 patients receiving MFD-HSCT. The conditioning regimen included a combination of Cy, Flu, TTP, and Bu, with a pre-transplant dose of Azat and HU on days 45 to 11. The two groups did not differ significantly in 3-year OS (MFD-HSCT vs. MUD-PBSC: 90.0% vs. 92.3%), TFS (83.3% vs. 90.4%), TRM (7.7% vs. 10.0%), cumulative graft failure incidence (6.9% vs. 1.9%), or grade III–IV acute GVHD (3.6% vs. 9.6%) [47]. Recently, MUD-PBSC using a protocol of PTI with HU, Flu, TTP, and Cy led to 100% OS and TFS, with only one patient developing GVDH, although the follow-up was short [63].

Previous studies reporting transplants from MFD-CB were associated with reduced GVHD risk but higher graft failure rates, ranging from 14% to 55.5% [19,30,31,39,59]. A subsequent study used CB- or BM-MFD and RIC involving alemtuzumab (Alem), Flu, and melphalan (Mel); this regimen yielded high OS and TFS in the β-thalassemia patient group [55]. The implementation of a Bu–Cy-based regimen with Flu, TTP, TBI, or TLI did not improve outcomes in a series of 35 unrelated CB-HSCT (UCB-HSCT), with graft failure and TRM rates of 57.1% and 14.2%, respectively, as well as reduced OS (62%) and TFS (21%) [45]. Increasing the stem-cell dose was associated with better HSCT outcomes, specifically 5-year OS of 88.3% and TFS of 73.9%, but acute and chronic GVHD rates were high [46]. More recently, similar results were reported for UCB-HSCT, although with even better OS of 100% [56]. However, a small group of young patients treated with an HU, Flu, and TTP protocol presented with different grades of GVHD severity, although no graft failure or TMR was reported [53].

Since the probability of finding a suitable MUD is conditioned by the patient’s ethnic background and HLA-genotype frequency, alternative donor source transplants have been considered, including partially mismatched related donors. The first study that collected transplants from mismatched relative donors was published in 2000 and adopted a Bu–Cy with ATG and variable TBI or TLI doses. Overall OS and TFS were 65% and 21%, respectively, with high rates of graft failure (55%), acute GVHD (47.3%), and chronic GVHD (37.5%) [17]. Protocol 26.1 was subsequently implemented in 16 β-TM patients receiving BM-HSCT from mismatched relative donors. A pretransplant treatment using HU (30 mg/kg/day), Azat (3 mg/kg/day), and Flu (30 mg/m^2^/day) was followed by intravenous Bu (3.5 mg/kg/day), TTP (10 mg/kg), Cy (50 mg/kg/day), and ATG (10 mg/kg). This regimen did not differ significantly in TMR (6% vs. 8%), graft failure (0% vs. 12%), OS (94% vs. 92%), or TFS (94% vs. 82%) from a cohort of 66 MSD-HSCT [52]. Recent studies have reported improvements in OS and TFS after myeloablative or RIC regimens were supplemented with PBSC-haplo-HSCT plus T-cell depletion and PTI. Unfortunately, graft failure and GVHD incidence was not negligible [44,57].

## 4. Discussion

This scoping review aimed to investigate most of the studies that described HSCT in β-thalassemia and analyze the evolution of conditioning regimes in this non-malignant disease, particularly with respect to progressive survival improvement over the past 30 years (Figure 2). We also considered several factors influencing the modulation of conditioning regimes (Figure 3). Our analysis revealed that Bu and Cy are the two most commonly used drugs in myeloablative conditioning regimens. The choice of two powerful alkylating agents was associated with the specific characteristics of β-thalassemia: a hyperplastic bone marrow and frequent allosensitization from multiple previous transfusions that require a strong approach to minimize graft failure risk [66]. Modulation in dosage and more attention to bioavailability has improved outcomes in patients with advanced disease [42]. The best results were obtained using BM stem cells from an MFD, with OS and TFS probabilities of over 90% and 80%, respectively [67]. However, differences between patient groups have been reported since the early 1990s, with Pesaro-risk class 3 and HR class 3 receiving more attention [11]. Earlier modulation methods involved reducing the Cy dose to decrease toxicity [13]. Since then, several protocols have been developed to minimize graft failure and GVHD, including early debulking of bone marrow hyperplasia and exploiting the highly immunosuppressive action of Flu [24,54]. Likewise, the new agent Treo was adopted because of its lower toxicity profile and VOD risk (classically linked to iron-mediated liver damage), with multiple studies demonstrating the drug’s ability to reduce the likelihood of GVHD [35,48,57].

Given that only 20–25% of patients have a potential sibling donor, we recommend MUD transplants, coupled with protocol modulation to reduce GVHD risk. The inclusion of Flu and TTP, as well as the introduction of drugs such as Alem or ATG, has successfully reduced GVHD in all cases of HLA disparity or non-related donors [17,20,25,30,45,46,47,51,53]. Although immunosuppressive therapy is the basic approach to prevent GVHD, careful donor selection can significantly improve clinical HSCT outcomes. Indeed, we now have a greater, more nuanced understanding of the different immunogenic and genetic factors implicated in HSCT. More attention is being given to HLA-DP classes, permitting and non-permitting mismatches, leading to research suggesting that immunogenetic factors should guide mismatch donor choice, while balancing host-versus-graft (HVG) and GVHD risk [68,69]. In addition, recent studies have shown that killer cell immunoglobulin-like receptors (KIRs) play a vital role in preventing GVHD [70].

Taken together, the outcomes of unrelated HSCT do not differ much from those obtained with a compatible sibling, provided that donor selection is stringent and protocols or drug dosage are adjusted accordingly [22,25,27]. Changes in conditioning regimes have evolved with the expansion of alternative donor stem cell sources. An elevated graft failure rate in CB-HSCT indicates the need for modification of pretransplant regimens or modulation of GVHD prophylaxis [19,30,31,39,45,46,47,53,55,56,58,59,62]. Further advancements are also required in PBSC-HSCT, and novel approaches with ex vivo graft manipulation of lymphocytes may be the way forward [44,57,71].

## 5. Conclusions

In our scoping review, we examined the evolution of conditioning regimens for β-thalassemia patients and highlighted the importance of patient characteristics such as age, Pesaro risk class, and availability of suitable donors for a successful HSCT. Over the years, conditioning regimens have been adapted to overcome patient-related limitations; increase availability of MUD, PBSC, and CB transplants for patients without a related donor; and minimize adverse outcomes (i.e., graft failure and GVDH).

Supportive care is crucial to preparing patients for transplantation. Iron chelation, blood transfusion support, and management of iron-related toxicities are constantly being improved and should be initiated as soon as possible during treatment [2]. Fortunately, the transplantation procedure has been extensively studied and successfully applied for the last four decades. Moreover, genetic therapies are becoming available in clinical trial settings, increasing the likelihood of further improvement in life expectancy [72].

However, this positive projection is less valid for developing countries that often experience geopolitical instability. In such regions, β-thalassemic patients lack access to supportive care and only have access to transplantation via humanitarian projects because highly specialized medical facilities are unavailable. Patients from low-income countries, where the incidence of the disease is higher, often fall into high-risk classes, with lower OS and TFS as well as greater TRM and graft failure.

In conclusion, our review demonstrates that better conditioning protocols lead to safer transplants and improved curative options for all patients with β-thalassemia.

## Figures and Tables

**Figure 1 jcm-11-00907-f001:**
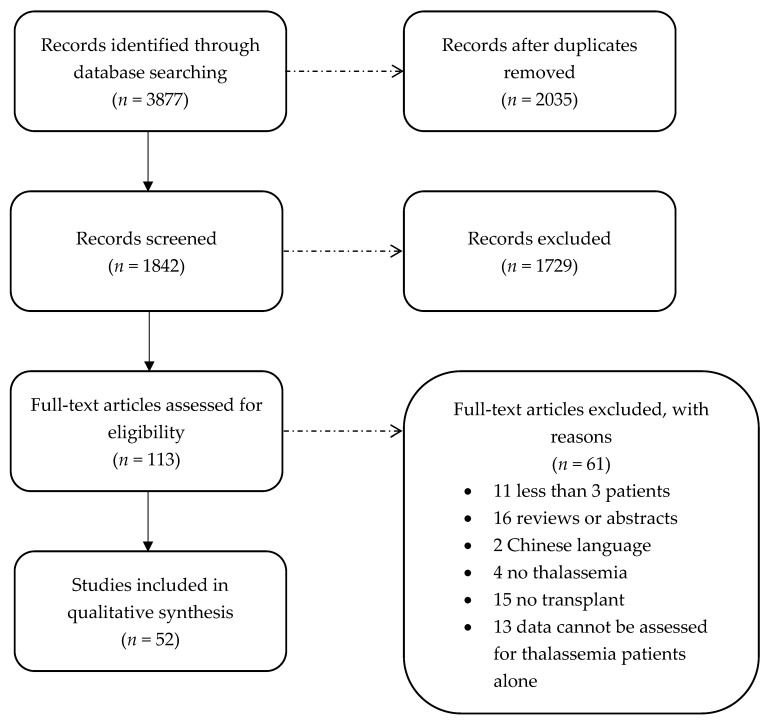
Prisma flow chart.

**Figure 2 jcm-11-00907-f002:**
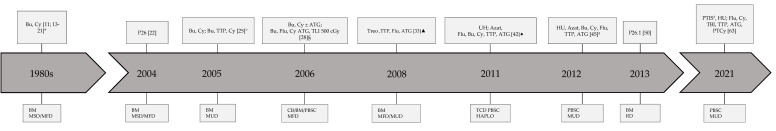
Notable studies in the history of β thalassemia HSCT. * First Bu, Cy based regimens; ° MUD HSCT more easily found in common practice; § CB HSCT more easily found in common practice; ▲ Bu is replaced by Treo in many regimens; ✦ HAPLO HSCT more easily found in common practice; ª PBSC stem cell source more easily used in common practice both for MUD and MFD.

**Figure 3 jcm-11-00907-f003:**
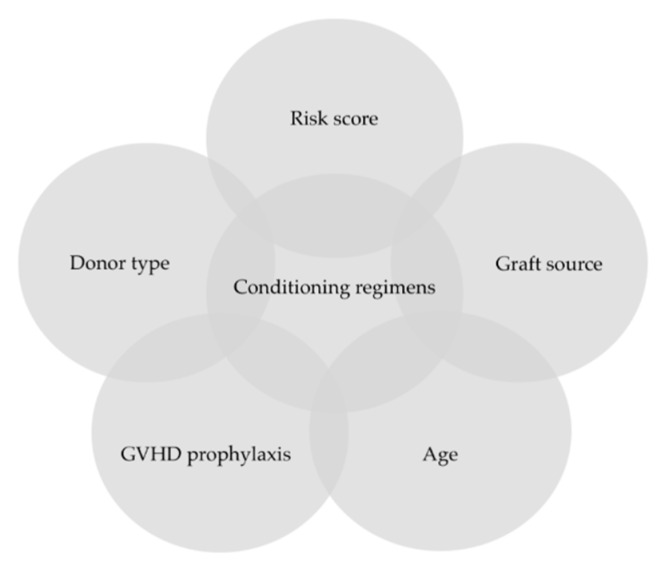
Factors influencing the modulation of conditioning regimes in β thalassemia HSCT.

**Table 1 jcm-11-00907-t001:** Principal studies that described protocol regimens in β-thalassemia hematopoietic stem cell transplantation. Bu, busulfan; Cy, cyclophosphamide; TTP, thiothepa; Treo, treosulphan; PTCy, post-transplant Cy; Flu, fludarabine; Azat, azathioprine; ATG, anti-thymocyte globulin; TG, thymoglobulin; CSA, ciclosporin; TCM, tacrolimus; SIL, silorimus; ALE, alemtuzumab; DEX, dexamethasone; MTP, methylprednisolone; MTX, methotrexate; MEL, melphalan; MFM, mycophenolate mofetil; TBI, total body irradiation; TLI, total lymphoid irradiation; BM, bone marrow; PBSC, peripheral blood stem cells; CB, cord blood; TCD, T cell depletion; MFD, matched family donor; MUD, matched unrelated donor; MSD, matched sibling donor; NA, not applicable; IP, interstitial pneumonitis; IH, intracranial hemorrhage; S, septicemia; E, encephalitis; CLS, capillary leak syndrome; VOD, veno-occlusive disease; HF, hepatic failure; CHF, congestive heart failure; AHA, autoimmune hemolytic anemia; HC, hemorrhagic cystitis; Tx, transplant; III^C^, Class 3 very high risk, d, day.

Author	Year	Country	N	Median Age Years	Sex M:F	Risk Score	Conditioning Regimen	GVHD Prophylaxis	Graft Source	Donor Type	aGVHD I–II/III–IV %	cGVHD %	TRM %	GF %	Other Complications	OS %	EFS %	FU M
Or [15]	1988	Israel	10	2	6:4	NA	TLI 200 cGy/dx4, BU 4 mg/kg/dx4, Cy 50 mg/kg/dx4	CSA	TCD BM	MSD, MFD	0	0	30	10	IH, IP	70	90	30
Lucarelli [11]	1990	Italy	116	6.5/10.5/12	NA	I-39/II-36/III-24	Bu 3.5 mg/kg/dx4, Cy 50 mg/kg/dx4	CSA, MTX, Cy	BM	MFD	4.3	0.8	5.1/19.4/37.5	0/8.3/16	CMV, ARDS	94/80/61	94/77/53	12
Lin [16]	1998	Taiwan	38	7	NA	NA	Bu 4 mg/kg/dx4, Cy 200 mg/kg ± TBI or TLI	CSA, MTX	NA	MFD, MUD	3/3	3	25	5.2	CMV, IP, IH, CHF	79	45	82
Suvatte [19]	1998	Thailand	35	NA	NA	NA	Bu 3.5 mg/kg/dx4, Cy 50 mg/kg/dx4; Bu 150 mg/m^2^/dx4, Cy 50 mg/kg/dx4	NA	BM, CB	MFD, MSD	0/3	0	3	14.3	IH, S	85.7	77.2	42
Lucarelli [13]	1998	Italy	393	<17	NA	I-121/II-272	Bu 3.5 mg/kg/dx4, Cy 50 mg/kg/dx4	CSA	BM	MFD	NA	NA	5/15	5/4	NA	95/85	90/81	NA
Lucarelli [13]	1998	Italy	125	<17	NA	III	Bu 3.5 mg/kg/dx4, Cy 30–40 mg/kg/dx4	CSA, MTX	BM	MFD	NA	NA	19	33	NA	78	54	NA
Rosales [20]	1999	Israel	14	NA	NA	NA	Bu 4 mg/kg/dx4, TTP 5 mg/kg/dx2, Cy 60 mg/kg/dx2	ALE, CSA	NA	NA	7/7	0	7.14	7.14	NA	92.8	NA	60
Gaziev [17]	2000	Italy	29	6	14:15	I-6/II-III 23	Bu 2–4 mg/kg/dx4, Cy 30–50 mg/dx4 ± TBI 300 400 cGy ± TLI ± ATG	CSA-MTX	NA	MFD, HAPLO	47.3	37.5	34	55	CMV	65	21	90
Khojasteh [21]	2001	Iran	63	11	37:26	II-8/III-55	Bu 3.75 mg/kg/dx4, Cy 50 mg/kg/dx4, ATG 40 mg/kg	CSA, PND	BM	NA	3/3	14	6.3	7.9	S	75	79.3	132
La Nasa [22]	2002	Italy	32	14	21:11	I-4/II-11/III-17	Bu 4 mg/kg/dx4, Cy 30–50 mg/kg/dx4; Bu 4 mg/kg/dx4, TTP 10 mg/kg, Cy 30–50 mg/kg/dx4	CSA, MTX	BM	MUD	41	25	19	12.5	VOD, CMV, IH, HF	79	66	30
Lawson [23]	2003	UK	55	6.4	26:29	I-17/II-27/III-11	Bu 3.5–4 mg/kg/dx4, Cy 30–50 mg/kg/dx4, ± FLU ± ATG	CSA, MTX	NA	MFD	84/15	16	5.4	13.2	VOD, CMV	94.5	81.8	75
Sodani [24]	2004	Italy	33	16	16:17	III	P26	CSA, PDN, MTX, CY	BM	MSD	9/0	NA	6	6	CMV	93	85	36
La Nasa [25]	2005	Italy	68	15	33:35	I-14/II-16/III-38	Bu 4 mg/kg/dx4, Cy 50 mg/kg/dx4; Bu 4 mg/kg/dx4, TTP 10 mg/kg, Cy 50 mg/kg/dx4; Bu 4 mg/kg/dx4, TTP 10 mg/kg, Flu 40 mg/m^2^/dx4	CSA, MTX	BM	MUD	40	18	20	13	NA	79.3	65.8	40
Chandy [26]	2005	India	47	6.74	31:16	II-21/III-25	Bu 150 mg/m^2^/dx4, Cy 50 mg/kg/dx4	CSA, MTX	NA	NA	30/19	NA	32	4	VOD, HC	68	68	63
Chandy [26]	2005	India	47	7.55	30:17	II-22/III-25	Bu 4 mg/kg/dx4, Cy 50 mg/kg/dx4, ATG 30 mg/kg/dx3	CSA, MTX	NA	NA	30/19	NA	28	9	VOD, HC	47	64	52
La Nasa [27]	2005	Italy	27	22	15:12	NA	Bu 3.5 mg/kg/dx4,Cy30–40 mg/kg/dx4; Bu 3.5 mg/kg/dx4 -TT 10 mg/kg- Cy 30–40 mg/kg/dx4	CSA-MTX ± ATG	BM	MUD	37	26	30	3.7	VOD CHF, CMV, IP, HF	70	70	43
Sauer [28]	2005	Germany	5	11.5	3:2	II-4/III-1	Bu 3.5mg/kg/dx4, Flu 30 mg/m^2^/dx6 ± ATG	CSA, MTX	BM	MSD	0	0	0	0	VOD	100	100	25
Elhasid [29]	2006	Israel	6	4.5	NA	I	Bu 4 mg/kg/dx4, Cy 30 mg/kg/dx4, Flu 40mg/m^2^/dx5, ATG 5 mg/kg/dx5	NONE	TCD PBSC	MSD	0	17	0	0	VOD, CLS, AHA	100	100	39.8
Hongeng [30]	2006	Thailand	28	7.2	13:15	I-15/II-III13	Bu 4 mg/kg/dx4, Cy 50 mg/kg/dx4 ± ATG 40 mg/kg; Bu 8 mg/kg, Flu 175 mg/m^2^, Cy 50 mg/Kg/dx4, ATG 20 mg/kg, TLI 500 cGy	CSA, TCM, MTX, MTP	BM, PBSC, CB	MFD	32/11	14	14.3	14	HC, S, AHA, VOD	92	82	51
Hongeng [30]	2006	Thailand	21	4	14:7	I-13/II-III18	Bu 4 mg/kg/dx4, Cy 50 mg/kg/dx4 ± ATG40 mg/kg	CSA, TCM MTX	BM	MUD	43/14	14	7.1	14	HC, S, VOD	82	71	35
Smythe [31]	2007	UK	7	5,8	NA	NA	Bu, Cy	NA	CB	MSD	0	0	0	28,5	NA	100	NA	45
Ullah [32]	2007	Pakistan	40	4	28:12	I-25/II-10/III-5	Bu 3.5 mg/kg/dx4, Cy 50 mg/kg/dx4; P26 in III	CSA, PND	BM, PBSC	MSD	40	13	20	12.5	VOD, HC, S, TB, CMV	80	72.5	48.9
La Nasa [33]	2007	Italy	45	33	25:20	I-14/II-18/III-13	Bu 3.5 mg/kg/dx4-TTP 10mg/kg-Cy 50 mg/kg dx4 or Cy 60 mg/kg/dx2; Bu 3.5 mg/kg/dx4, TT10 mg/kg Flu 40 mg/m^2^/dx4	CSA-MTX	BM	MUD	44	NA	13.3	15.6	NA	86.7	71.4	55
La Nasa [33]	2007	Italy	53	12	NA	I-18/II-21/III-14	Bu 3.5mg/kg/dx4, Cy 30–50mg/kg/dx4; Bu 3.5 mg/kg/dx4, TTP10 mg/kg, Flu 40 mg/m^2^/dx4	CSA-MTX	BM	MUD	30	NA	11.3	15.1	CMV, IP	88.7	73.6	NA
Bazarbachi [34]	2008	Lebanon	10	5	4:6	III	Bu/Cy/ ± ATG	NA	NA	NA	10/10	NA	10	10	VOD, CMV, S	89	67	65
Bernardo [35]	2008	Italy	20	13	14:6	I-7/II-4/III-9	Treo 14 g/m^2^/dx3, TTP 8 mg/kg, Flu 40 mg/m^2^/dx4, ATG 10 mg/kg/dx3	CSA, MTX	BM	MFD, MUD	10/5	5	5	10	NA	95	85	20
Ghavamzadeh [36]	2008	Iran	96	6	49:47	I-40/II-56	Bu 4 mg/kg/dx4, Cy 50 mg/dx4	CSA, MTX	BM	MFD	35	18	11	3.1	NA	89	76	29
Ghavamzadeh [36]	2008	Iran	87	5	57:30	I-48/II-39	Bu 4 mg/kg/dx4, Cy 50 mg/dx4	CSA, MTX	PBSC	MFD	72	48	17	0	NA	83	76	60
Dennison [37]	2008	Oman	41	9	NA	II-23	Bu–Cy 50 mg/kg/dx4 ± ATG; Bu 3.5mg/kg/dx4, Flu 30 mg/m^2^/dx6 ± ATG	NA	BM CB-BM	NA	19	5	10	7	NA	88	88	72
Di Bartolomeo [38]	2008	Italy	115	9	57:58	NA	Bu 3.5–4mg/kg/dx4, Cy 50mg/kg/dx4	CSA, MTX	BM	MFD	43	20	8.7	6.7	E, CHF	89.2	85.7	180
Ghavamzadeh [39]	2009	Iran	9	6.7	NA	I-4/II-5	Bu 3.5–4 mg/kg/dx4, Cy 50 mg/dx4	CSA, MTX	CB	MSD	0	0	11.1	55.5	NONE	88.9	33.3	NA
Kumar [40]	2009	India	4	6	4:0	II-1/III-3	Bu 4 mg/kg/dx4, Cy 50 mg/kg/dx4, ATG 30 mg/kg	CY, MTX	BM, PBSC	NA	NA	NA	0	NA	NA	100	NA	18
Ramzi [41]	2009	Iran	155	9.5	NA	I-43/II-53/III-59	Bu 4 mg/kg/dx4, Cy 30–50 mg/kg/dx4, ATG 40 mg/kg	CSA, MTX	BM, PBSC	NA	NA	NA	14.8	12.9	NA	85.1	74.1	97.2
Chiesa [42]	2010	Italy	53	8	29:24	I-2/II-26/III-25	Bu, Cy 50 mg/kg/dx4 ± TTP 10 mg/kg; P26, ^¥^Bu, and Cy 4 mg/kg/dx4	CSA	BM	MSD MFD	7	4	4	21	VOD AKI	96	79	18.5
Caocci [43]	2011	Italy	28	10	17:11	II-4/III-24	Treo, TTP, Flu; Bu Cy Bu, TTP, Cy	CSA -MTX	NA	MFD, HAPLO, MUD	21/14	18	10.9	14.3	NA	89.3	78.6	24
Sodani [44]	2011	Italy	31	NA	NA	NA	HU 60 mg/kg/dx50; Azat 3 mg/kg/dx48, Flu 30 mg/m^2^/dx5, Bu 3 mg/kg/dx4, Cy 50 mg/kg/dx4, TTP 10 mg/kg, ATG 12.5 mg/kg/dx4	CSA	TCD PBSC	HAPLO	0	0	6.5	22.5	CMV, EBV	93.5	71	NA
Ruggeri [45]	2011	France	35	NA	20:15	NA	Bu 8 mg/kg or Bu 6.4 mg/kg, Cy, ATG; other regimens Bu, Cy ± Flu ± TTP ± ATG ± TLI ± TBI	CSA, MFM, MTX, ALE	CB	MUD	NA	NA	14.2	57.1	VOD, MOF, ARDS	62	21	21
Jaing [46]	2012	Taiwan	35	5.5	16:19	NA	Bu 3.5 mg/kg/dx4, Cy 50 mg/kg/dx4, ATG 30 mg/Kg/dx4 or thymoglobulin 3 mg/Kg/dx4	CSA, MPN	CB	MUD	51/46	40	11.4	17.1	IH, S, IP	88.3	73.9	36
Li [47]	2012	China	30	6	20:10	II	HU 30 mg/kg/d, Azat 3 mg/kg/d, Bu 2.8–4.4 mg/kg/dx 4, Cy 60 mg/kg/dx2, Flu 40 mg/m^2^/dx4, TTP 10 mg/kg, ATG 15–30 mg/Kg/dx4 or TG 2.5 mg/Kg/dx4	CSA, MFM, MTX	BM, CB-BM	MFD	0/3.6	0	10	6.9	CMV, VOD	90	83.3	24
Li [47]	2012	China	52	6	36:16	II	HU 30 mg/kg/d, Azat 3 mg/kg/d, BU 2.8–4.4 mg/kg/dx 4, Cy 60 mg/kg/dx2, Flu 40 mg/m^2^/dx4, TTP 10 mg/kg, ATG 15–30 mg/Kg/dx4 or TG 2.5 mg/Kg/dx4	CSA, MFM, MTX	PBSC	MUD	0/9.6	0	7.7	1.9	CMV, VOD	92.3	90.4	24
Bernardo [48]	2012	Italy	60	7	32:28	I-27/II-17/III-4	Treo 14 g/m^2^/dx3, TTP 8 mg/kg, Flu 40mg/m^2^/dx4, ATG 10 mg/kg/dx3	CSA, MTX	BM, PBSC, CB	MFD, MUD	7/7	2	6.6	9	IP	93	84	36
Goussetis [49]	2012	Greece	75	7	37:38	I-16/II-38/III-17	Bu 3,5–4 mg/kg/dx4, Cy 37.5–50 mg/kg/dx4, ± Flu 25 mg/m^2^/dx4, ATG	CSA, MTX	BM, BM + CB, PBSC	MFD	9/4	13	4	4	NA	96	92	108
Choudhary [50]	2013	India	28	9,6	15:13	II-7/III-21	Treo 14 g/m^2^/dx3, TTP 8 mg/kg, Flu 40mg/m^2^/dx4	CSA, MTX	NA	MFD	4	2	21.4	7.14	VOD	78.5	71.4	13
Mathews [51]	2013	India	193	>11	118:75	III-139/III^c^-54	Bu 4 mg/kg/dx4, Cy 50 mg/kg/dx4, ATG 30 mg/kg/dx3; Bu 150 mg/m^2^/dx4, Cy 50 mg/kg/dx4	CSA, MTX	BM, PBSC	MFD, MUD	44	18	28	12	IH, VOD, HC,	63.6/39.4	57.3/32.4	42
Mathews [51]	2013	India	74	>11	46:28	III-50/III^c^-24	TTP 8 mg/kg, Treo 14 g/m^2^/dx3,Flu 30 mg/m^2^/dx4	CSA, MTX	BM, PBSC	MFD, MUD	35	11	12	8	VOD	87.4/86.6	78.8/77.8	42
Gaziev [52]	2013	Italy	16	9.6	10:6	I-5/II-5/III-10	P26.1	CSA, MTX, PND, CY	BM	RD	19/13	13	6	0	HC	94	94	72
Gaziev [52]	2013	Italy	66	10	37:29	I-0/II-31/III-35	Bu 3.5 mg/kg/dx4, Cy 50 mg/dx4 ± TTP; Bu 3.5 mg/kg/dx4, Cy 50 mg/dx4; HU30 mg/kg/d, Azat3 mg/kg/day, Flu20 mg/m^2^, Bu 3.5 mg/kg/dx4, Cy 22.5–40 mg/dx4 ± TTP	CSA, MTX, PND, CY	BM	MSD	36/7	11	8	12	VOD, HC	92	82	80
Hussein [18]	2013	Jordan	44	8	20:24	I-7/II-24/III-13	Bu 5–4 mg/kg/dx4, Cy 50 mg/kg/dx4; Bu 2 mg/kg/dx5, Flu 35 mg/m^2^/dx5, TLI 500 cGy	CSA, MTX, MFM	BM, PBSC	MFD	32	16	2.2	11,3	NA	97.8	86.4	64
Parikh [53]	2014	US	4	3.3	3:1	NA	HU 30 mg/kg/dx12, Flu 30 mg/m^2^/dx5, TTP 200 mg/m^2^ × 1	TCM, MFM, ALE	CB	MUD	50/50	25	0	0	AHA, EBV	100	NA	19.7
Anurathapan [54]	2014	Thailand/US	76	8	44:32	III	Bu 1 mg/kg/dx4 or Bu 0.95–1.2 mg/kg/dx 4, Cy 50 mg/kg/dx4; Bu 1 mg/kg/dx4 or Bu 0.95–1.2 mg/kg/dx 4, Cy 50 mg/kg/dx4, Flu 30 mg/m^2^/dx6, ATG 10 mg/Kg/dx4	CSA, TCM, MTX	BM, PBSC	MFD, MUD	17/7	11	7	8	HC, S, AHA, VOD, CMV	95	88	114
Anurathapan [54]	2014	Thailand/US	22	17	9:13	III^c^	PTIS^b^x2, Flu 35 mg/m^2^/dx6, Bu 130 mg/m^2^/dx4, ATG 1.5 mg/kg/dx3	CSA, TCM, MFM	BM, PBSC	MFD, MUD	9/14	18	9	0	HC, VOD, CMV	90	93	36
King [55]	2015	US/Canada	9	NA	NA	NA	Flu 35–37.5 mg/m^2^/dx4, MEL 150 mg/m^2^	CSA, TCM, MTX, PND, MFM, ALE	BM, CB	MSD	NA	NA	0	NA	NA	100	100	41
Hussein [14]	2015	Jordan/US	29	13.9	14:15	III	Bu 4 mg/kg/dx2, Flu 35 mg/m^2^/dx5, ATG 30 mg/kg/dx5 in 7; ATG 2.5 mg/kg/dx3 in 22; TLI 500 cGy single dose	CSA, MFM	PBSC	MSD, MFD	11	8	0	20.6	VOD	100	NA	45.3
Shah [56]	2015	India	9	3.8	8:1	NA	Bu 4 mg/kg/dx 4, Cy 50 mg/kg/dx2, Flu 90 mg/kg/dx2, ATG 7.5 mg/Kg/dx3	CSA, MPN	CB	MUD	33/0	0	0	44	CMV	100	56	22.6
Anurathapan [57]	2016	Thailand	31	10.1	17:14	I-7/II-9/IIIc-15	PTISbx2, Flu 35 mg/m^2^/dx6, Bu 130 mg/m^2^/dx4, ATG 1.5 mg/kg/dx3	CY, TCM or SIL, MFM	TCD-PBSC	HAPLO	29/3.2	16.1	3.2	3.2	VOD, CMV	95	94	12
Zaidman [58]	2016	Israel	34	8	NA	I-7/II-6/III-21	P26; P26.1; BU 3.5 mg/kg/dx4, Cy 50 mg/kg/dx4;Bu 4 mg/kg/dx4, Cy 30 mg/kg/dx4, Flu 40 mg/m^2^/dx5, ATG 5 mg/kg/dx5	CSA, MTX, TTP, ATG	BM, PBSC, CB ± TCD	MFD, MUD	NA	NA	14.7	8.8	VOD, CLS	90.5	81.7	129
Gabr [59]	2017	Egypt	6	5.5	3:3	II	Bu 4 mg/kg/dx4, Cy 30 mg/kg/dx4, ATG11 mg/kg/dx4	DEX, MTP, CSA	PBSC, CB	MSD	17/0	NA	16.7	33.3	NA	83.3	50	24
Caocci [60]	2017	Italy	258	12	140:118	I-57/II-83/III-21	Bu Cy 30–50 mg/kg/dx4; Bu –Cy 30–50 mg/kg/dx4,TTP; Treo-TT-Flu, ATG	CSA, TCM, MPD, MTX	NA	MFD, MUD	23.6	12.9	13.8	6.9	VOD, IH	82.6	77.8	132
Park [61]	2018	Korea	15	6.2	6:9	NA	Bu 130 mg/m^2^/dx 4, Cy 60 mg/kg/dx2, ATG2.5mg/kg/dx3	CSA, MTX	BM, PBSC	NA	0/7	0	0	0	VOD	100	NA	27
Benakly [62]	2020	Algeria	47	7.6	NA	NA	Bu 3 mg/kg/dx4, Cy 30–50 mg/kg/dx4, ATG10mg/kg	CSA, MTX	BM, PBSC, CB	MUD	28	9	NA	14.8	NA	75.7	66.8	180
Kharya [63]	2021	India	4	6	1:2	III	PTIS^a^ x2 cycles, HU 20 mg/kg/dx50; F30 mg/m^2^/dx5, CY 14.5 mg/kg/dx2, TBI 2 Gyx1, TTP 10 mg/kg/dx1, ATG 1.5 mg/kg/dx3, PTCY dx2	PTCY, SIL, MFM	PBSC	MUD	25/0	0	0	0	NONE	100	100	10.2

P26 protocol26: hydroxyurea 30 mg/kg/d, azathioprine 3 mg/kg/d, fludarabine 20 mg/m^2^/dayx5, busulfan 3.5 mg/kg/dayx4, cyclophosphamide 40 mg/kg/dayx4, ATG; P26.1 protocol 26.1: hydroxyurea 30 mg/kg/day, azathioprine 3 mg/kg/day, fludarabine 30 mg/m^2^/dayx4, busulfan 3.5 mg/kg/dayx4, thiotepa 10 mg/kg, cyclophosphamide 50 mg/kg/dayx4, and antithymocyte globulin 10 mg/kg; PTIS^a^ pre-transplant immune suppression: fludarabine 150 mg/m^2^/dx5, ciclophosphamide 1 g/m^2^/dx1, dexamethasone 20 mg/m^2^/dx5; PTIS^b^ fludarabine 40 mg/m^2^/dayx5, dexamethasone 25 mg/m^2^/dayx5.

## Data Availability

Not applicable.

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
