# Peer review of "Conditioning Regimens in Patients with β-Thalassemia Who Underwent Hematopoietic Stem Cell Transplantation: A Scoping Review"

_jcm, 2022, doi:10.3390/jcm11040907_

Round 1

Reviewer 1 Report

page 31 , Beta-thalassemia major (β-thalassemia) is a common monogenic disease characterized by abnormal hemoglobin structure. is wrong. Thalassaimia is not an abnormal hemoglobin structure . please correct

page 98, Overall, 52 patients were included, patients or studies?

The results are long and It is difficult to be understanding from physician who is not working at a transplantation unit. Some tables may be useful .

Author Response

Response to Reviewer 1 Comments

Point 1: page 31 , Beta-thalassemia major (β-thalassemia) is a common monogenic disease characterized by abnormal hemoglobin structure. is wrong. Thalassaimia is not an abnormal hemoglobin structure . please correct

Response 1: Thank you for pointing that out. We have revised the text to read as follow: Beta-thalassemia major (β-thalassemia) is a common monogenic disease characterized by reduced or no production of hemoglobin. 

Point 2: page 98, Overall, 52 patients were included, patients or studies?

Response 2: There was a mistake in the text. Please find in the revised text the change of: studies

Point 3: The results are long and It is difficult to be understanding from physician who is not working at a transplantation unit. Some tables may be useful .

Response 2: Thank you for your feedback. You can find attached two figures that simplify the complexity of the topic.                                                

Reviewer 2 Report

The review article presents a conceptual analysis of conditioning regimens used for hematopoietic stem cell transplantation (HSCT) in β-thalassemia major patients. The basic Bu/Cy conditioning regimen was applied in most studies published worldwide. Graft failure and GVHD are the main limiting factors for HSCT success in this disorder. The authors have selected 52 relevant studies from the total of 3877 publications. Gradual improvement of survival rates was traced from the beginning of HSCT in thalassemia, due to optimized donor selection and introduction of reduced-intensity conditioning (RIC) protocols. In this context, pre-transplant immunosuppression is considered a feasible option in the high-risk cases. Moreover, the authors discuss usage of non-marrow stem cell sources (peripheral, or cord blood stem cells), despite higher risks of graft failure or severe GVHD. In this respect, HSCT from haploidentical donors also becomes a plausible option. The important role of  conditioning regimens in thalassemia patients is declared, but still cannot be assessed, taking into account limited number of cases treated under similar donor selection, pre-conditioning treatment, or GVHD prophylaxis, thus causing rather different outcomes worldwide.

Still it lacks statistical evaluation, e.g., forest plot graphs for different conditioning modes, due to heterogeneity of additional treatment factors affecting outcomes in multiple studies. However, general conclusions are presented in objective manner and are of sufficient clinical value.

Author Response

Response to Reviewer 2 Comments

Thank you for your feedback.

The lack of an appropriate statistical evaluation in this article, is obviously due to large heterogeneity. Nevertheless, the purpose of our work was to summarize the different conditioning regimes in thalassemia. In our opinion, a scoping review would have been better suited to this analysis.
